# Single-shot laser-driven neutron resonance spectroscopy for temperature profiling

Zechen Lan[1], Yasunobu Arikawa[1], Seyed Reza Mirfayzi[2], Alessio Morace[1], Takehito Hayakawa[3], Hirotaka Sato[4], Takashi Kamiyama[4], Tianyun Wei[1], Yuta Tatsumi[1], Mitsuo Koizumi[5], Yuki Abe[1,6], Shinsuke Fujioka[1], Kunioki Mima[1], Ryosuke Kodama[1] & Akifumi Yogo[1] ✉

The temperature measurement of material inside of an object is one of the key technologies for control of dynamical processes. For this purpose, various techniques such as laser-based thermography and phase-contrast imaging thermography have been studied. However, it is, in principle, impossible to measure the temperature of an element inside of an object using these techniques. One of the possible solutions is measurements of Doppler brooding effect in neutron resonance absorption (NRA). Here we present a method to measure the temperature of an element or an isotope inside of an object using NRA with a single neutron pulse of approximately 100 ns width provided from a high-power laser. We demonstrate temperature measurements of a tantalum (Ta) metallic foil heated from the room temperature up to 617 K. Although the neutron energy resolution is fluctuated from shot to shot, we obtain the temperature dependence of resonance Doppler broadening using a reference of a silver (Ag) foil kept to the room temperature. A free gas model well reproduces the results. This method enables element(isotope)-sensitive thermometry to detect the instantaneous temperature rise in dynamical processes.

The temperature measurement of material inside of an object is one of the key technologies for control of dynamical processes. For this purpose, various techniques such as phase-contrast imaging thermography[1] and laser-based thermography[2] have been studied. However, it is, in principle, impossible to measure the temperature of an element inside of an object using these techniques. One of the possible solutions is measurements of Doppler brooding effect in neutron resonance absorption (NRA)[3]. Neutrons can be used to obtain isotopic, thermodynamic, and structural information from materials because of their high transmittance even at relatively low energies of approximately meV or eV. The NRA is the process in which a neutron at an energy related to the excited states of the atomic nucleus just above the neutron binding energy is resonantly captured by the nucleus. Thus, the probability that neutrons interact with nuclei depends on the incident energy and atom identity[4]. Bethe[3] suggested the temperature dependence of the broadening of a resonance peak due to the Doppler effect, in which the incident energy of neutron to a nucleus changes due to the thermal motion of the nucleus. Thus, the neutron resonance spectroscopy (NRS) provides a possibility of detecting the temperature as well as the elemental composition and density inside a material non-destructively. The NRS has been predominantly implemented with accelerator-based neutron sources for material analysis[5–7], spatially resolved thermometry[5,8,9], and shock wave measurement[10,11]. In these methods, neutron energy spectra are measured by using the time-of-flight (TOF) method with a pulsed neutron beam. The TOF method is to obtain neutron energy by calculation with a known neutron flight distance and time. A pulsed neutron source is required for the TOF to ensure all the neutrons start at same time. Nevertheless,

---

[1]Institute of Laser Engineering, Osaka University, Suita 565-0871, Japan. [2]Tokamak Energy Ltd, Oxford OX14 4SD, UK. [3]Kansai Institute for Photon Science, National Institutes for Quantum Science and Technology, Kizugawa 619-0215, Japan. [4]Faculty of Engineering, Hokkaido University, Sapporo 060-8628, Japan. [5]Integrated Support Center for Nuclear Nonproliferation and Nuclear Security, Japan Atomic Energy Agency, Tokai 319-1195, Japan. [6]Graduate School of Engineering, Osaka University, Suita 565-0871, Japan. ✉e-mail: yogo.akifumi.ile@osaka-u.ac.jp

it is impractical to generate a neutron pulse with perfect simultaneity, resulting in a finite pulse duration that introduces measurement errors in the determination of the flight time. To obtain sufficient energy resolution, a long beamline which is typically a few tens of meters for accelerator-based neutron sources[12] must be set for TOF measurements. However, a long beamline reduces the flux of neutrons arriving at a detector due to the smaller solid angle of the detector. To obtain a neutron spectrum with sufficient statistics, neutron pulses must be integrated for several hours in typical neutron facilities. As a result, even if the pulse width and flux of accelerator-based neutron sources are stable, the energy resolution becomes lower than that of a single neutron pulse.

As an alternative source, laser-driven neutron sources (LDNS)[13–22] have attracted widespread attention for their compactness and short pulse performance. An LDNS utilizes laser-driven ion acceleration[23–27], which enables the acceleration of protons or deuterons up to a few tens of MeV. By placing a secondary target, typically beryllium (Be) or lithium (Li), in the beam path, a pulse of MeV-energy neutrons is generated by nuclear reactions between the target materials and laser-accelerated ions. A pulsed fast neutron beam can be generated by a LDNS of ~4 cm size within a pulse duration of ~1 ns. A currently highest neutron yield of ~$10^{11}$ has been achieved[15,19,20,22] for a single pulse. Furthermore, low-energy neutrons at eV ~meV region have been generated by employing neutron moderators made of hydrogen-rich materials at room temperature[22,28–30] or cryogenically cooled solid hydrogen[31]. The short pulse duration and miniature scale of an LDNS allow a smaller neutron moderator, resulting in a reduction of the neutron pulse expansion in the moderation process. At present, accelerator-based neutron sources are more stable than LDNS. However, when a single laser shot is used for measuring the NRA, the energy resolution originating from the short-term expansion of the initial neutron pulse is expected to be higher than that using an accelerator-based neutron source. Previous studies have demonstrated NRA using LDNS[30,32,33]. In our last work[22], we measured neutron resonances in the epithermal (several eV) region using a 1.8 m beamline after an LDNS, where one resonance spectrum was obtained with a single bunch of neutrons generated by a single pulse of the laser.

In this study, we report measurements of the Doppler broadening of a NRA generated with a single neutron pulse provided by laser. We demonstrate that the Doppler width of the resonance absorption of an atomic nucleus $^{181}$Ta increases as the square root of the sample temperature according to a theoretical model. Each Doppler spectrum is acquired from a single pulse of a laser. This result indicates the possibility that an LDNS may provide a real-time thermometry of a nuclide that probes the instantaneous temperature of dynamic objects.

## Results
### Single shot measurement of NRA
The experimental setup is shown in Fig. 1 and the details are described in METHODS. We obtained neutron energy spectra with the resonance absorption of the the Ag (thickness = 0.2 mm) and Ta (thickness = 0.1 mm) foils, where the Ta foil was heated up to $T$ = 361, 413, 474, 573, and 617 K but the Ag foil was kept to the room temperature for using as a reference. Figure 2a shows the signal of epithermal neutrons, which transmitted through the Ag and Ta foils at the room temperature, recorded by the $^6$Li-TOF detector. In the time region of 50–60 μs, two distinct troughs are seen in the continuous neutron signal. The flight time $t_{raw}$ measured by the detector is expressed by $t_{raw}=(t*D)(t)$, where $t$ is the flight time of neutrons that is deconvolved from the systematic pulse broadening $D(t)$ (see Methods). The neutron kinetic energy $E$ is obtained by $E=m(d/t)^2/2$, where $d$ = 1.78 m is the flight distance and $m$ is the mass of the neutron. The energies of the two dips correspond to the resonance peaks in Fig. 2b. The neutron absorption cross sections at 300 K are ~12,500 and 23,000 barns at 4.28 eV ($^{181}$Ta) and 5.19 eV ($^{109}$Ag), respectively.

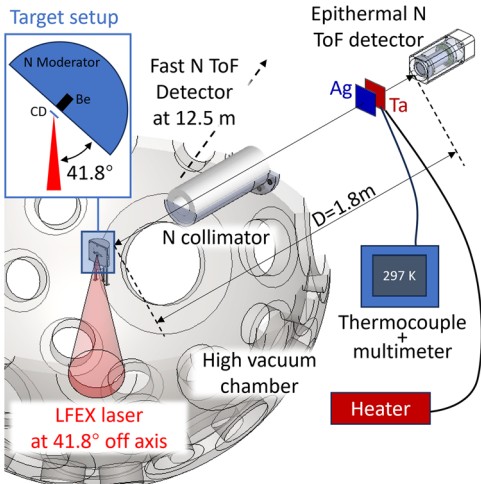

**Fig. 1 | The experimental setup of the laser-driven epithermal neutron generation and resonance absorption measurement using the TOF method.** The target setup shows spatial position relationship of the laser beam, CD foil, Be and neutron moderator. The laser-target chamber was kept at high vaccum during the experiment. The beamline setup including samples and detectors are shown in the figure.

In present analysis, we evaluate the background level for each laser shot using the reference foil (Ag) in the following procedure similar to black resonance[34]. The baseline [the blue dashed line in Fig. 2a] is obtained from a least-square fitting for the raw signals excluding the two resonance dips. The background (the green solid line) is obtained from the baseline by subtracting the depth of the resonance peak of $^{109}$Ag, because the Ag target is enough thick to absorb almost all neutrons around the resonance energy. From the difference between the raw signal and the baseline, the experimental absorption rate $R_{exp}(E)$ normalized to the difference between the baseline and the background is obtained as the blue dashed line in Fig. 2c. The absorption rate $R_0(E)$ for the Ag and Ta foils is theoretically analyzed by

$$R_0(E)=1-\prod_{Ag,Ta}\exp(-nl\sigma(E)),\qquad(1)$$

where $n$ is the volumetric number density, $l$ is the thickness of the targets and $\sigma(E)$ is the NRA cross section (JENDL4.0 data base[35]) at 300 K [Fig. 2b]. The black line in Fig. 2c shows the theoretical absorption rate when $t_{raw}=t$ without considering the effect of $D(t)$. $R_{exp}(E)$ exhibits two peaks at 4.3 and 5.2 eV which are also found in the analytical model $R_0(E)$. However, the detailed shape of the experimental peaks is not well reproduced by $R_0(E)$, especially for the thicker target (Ag), where the saturated absorption observed in $R_0(E)$ is not exhibited in $R_{exp}(E)$. This difference indicates the presence of another effect, $F(t)$, that causes further pulse broadening in addition to $D(t)$.

$F(t)$ is considered to originate from random neutron scattering in the beamline and involved in a Gaussian form. We determine $F(E)$ as a function of E by fitting the experimental absorption rate $R_{exp}(E)$ with the following equation:

$$R_1(E)=(R_0*F)(E).\qquad(2)$$

Fig. 3a shows $R_{exp}(E)$ (gray) fitted with the model $R_1(E)$ (red) by using an nonlinear-curve-fitting tool with a weighted region around resonances. The resonance peaks are well reproduced as the errors shown in the lower frame. $F(t)$ has a half-width of ~100–200 ns, which is sufficiently shorter than $D(t)$ (~0.5 μs). This result indicates that the advantage of

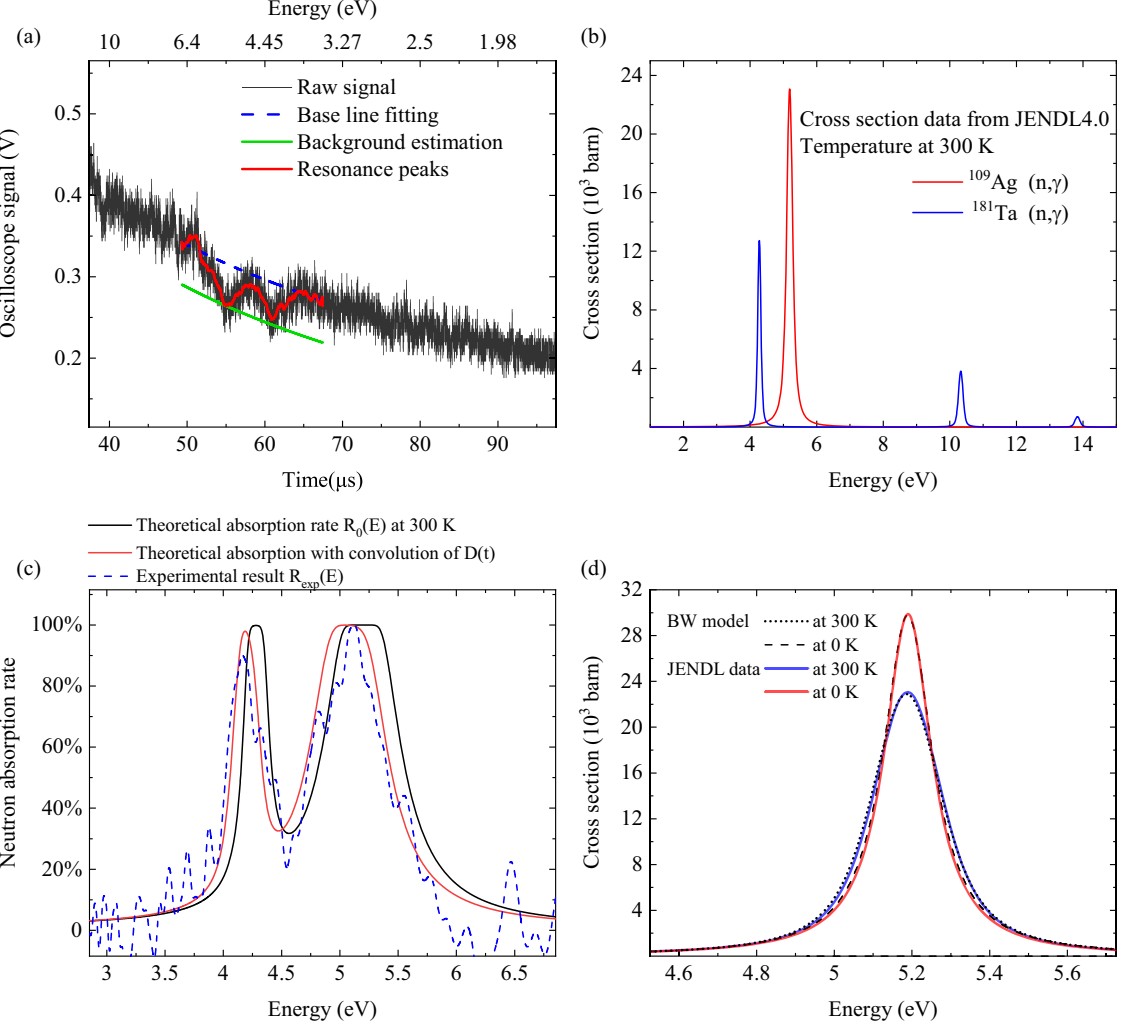

**Fig. 2 | Experimental neutron spectrum and analysis. a** The epithermal neutron signal. Two troughs can be checked in a continuous curve which should be exponential. **b** The neutron absorption cross sections of $^{109}$Ag and $^{181}$Ta at 300 K, obtained from JENDL4.0 data base[35]. **c** Experimental and theoretical neutron absorption. **d** JEDNL4.0 cross section data and $\sigma_{BW}(E)$ calculation of a $^{109}$Ag resonance at 5.19 eV.

the miniature size of the moderator is not offset by the pulse broadening caused along the beamline.

Although the two resonance peaks are close to each other in the present experiment, this overlapping region was already considered in the FWHM analysis as explained later. When we chose the reference material whose resonance energy is enough far from the resonance energy of the material to be measured, the uncertainty caused from the overlapping between the two resonance energies becomes negligibly small.

**Temperature dependence on resonance Doppler broadening**
We obtained temperature dependence on the NRA of the Ta foil. The Ta foil was heated to $T$ = 361, 413, 474, 573, and 617 K, whereas the Ag foil was kept to the room temperature for using as a reference. Every temperature case was measured in single laser pulse.

To analyse the temperature dependence of the resonance absorption, we introduce a Breit-Wigner (BW) single-level formula[4], as seen in refs. 10,36, for the neutron absorption cross section at 0 K:

$$\sigma_{BW}(E) = \pi \lambda g_j \frac{\Gamma_n \Gamma_\gamma}{(E - E_r)^2 + (\Gamma_n + \Gamma_\gamma)^2/4}, \quad (3)$$

where $E_r$ is the resonance energy, $E$ is the kinetic energy of the incident neutron, $\lambda$ is the de Broglie wavelength of incident neutron (divided by $2\pi$), and $g_j$ is the statistical factor determined by the angular momentum. $\Gamma_n$ and $\Gamma_\gamma$ represent the resonance width for the neutron and decay width for $\gamma$-ray, respectively. For the $^{181}$Ta resonance at $E_r$ = 4.28 eV, $\Gamma_n$ = 1.74 meV and $\Gamma_\gamma$ = 55 meV; For the $^{109}$Ag resonance at $E(r)$ = 5.19 eV, $\Gamma_n$ = 8.34 meV and $\Gamma_\gamma$ = 136 meV[35]. We have confirmed the $\sigma_{BW}(E)$ formula by fitting it to the latest analytical result of JENDL4.0 data base[35] for 0 K and 300 K [Fig. 2d]. The averaged differences on cross sections were evaluated as <10 barns.

When the thermal motion of target nuclei is sufficiently slower than the incident neutron, the Doppler broadening effect can be well approximated by a Gaussian function[8,10]. Therefore, we developed an analytical cross-section $\sigma_T(E, T)$ for the NRA involving the target temperature as follows

$$\sigma_T(E, T) = A \times \sigma_{BW}(E')* \exp\left(-\frac{(E' - E_r)^2}{2\Gamma_D^2(T)}\right), \quad (4)$$

where $A$ is a fitting parameter and $E'$ is the relative neutron energy for integration variable. $\sigma_{BW}$ and the following Doppler broadening term are convoluted in the energy axis. The Doppler width $\Gamma_D(T)$ broadens

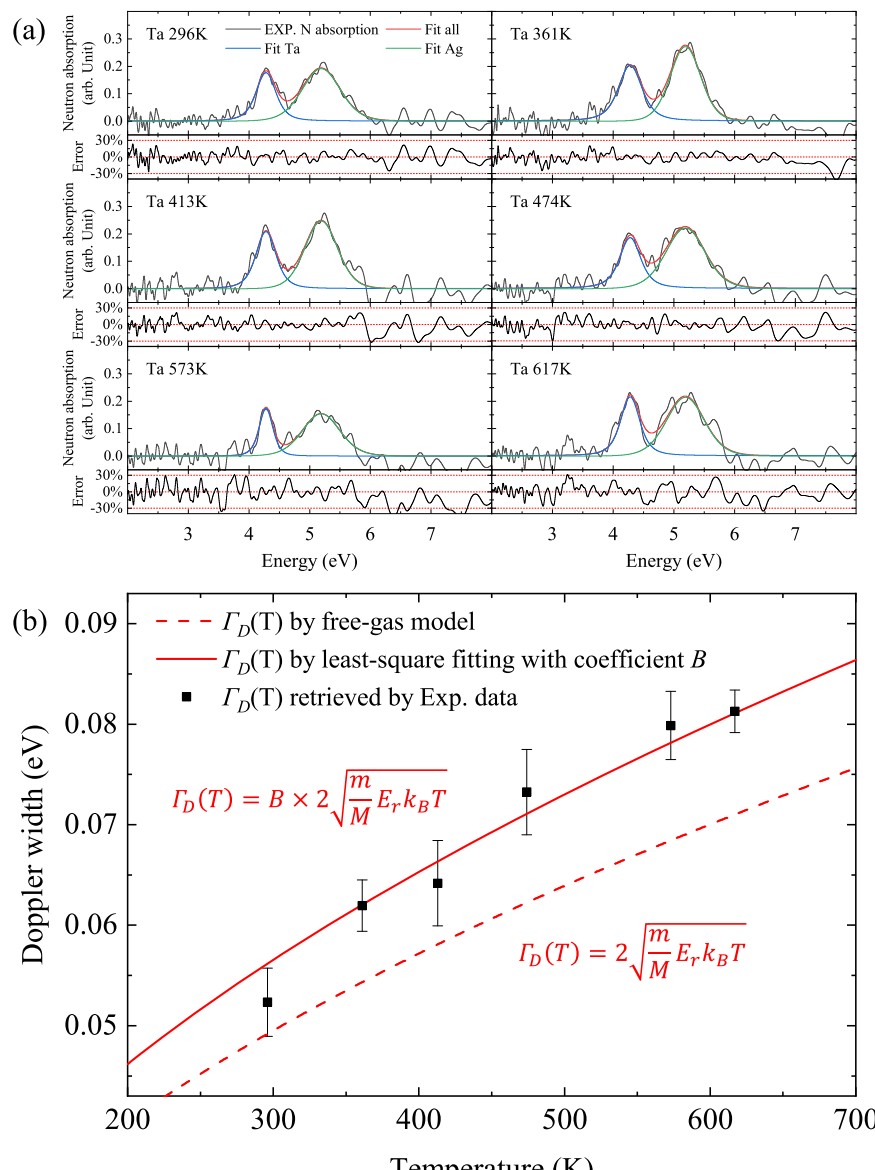

**Fig. 3 | Analysis of temperature dependent neutron resonance width.**
**a** Experimental neutron absorption results and model fitting by $R_T(E, T)$ [Eq. (5)]. The temperature of Ag was kept at 296 K and Ta was heated to $T$ = 297, 361, 413, 474, 573, and 617 K. **b** Theoretical Doppler width and experimental results. The error bars of Doppler width depend on the fitting error and the noise level of the original signal. The temperature of each data point was measured by the thermocouple in the experiment.

with the temperature $T$. Then, the temperature-dependent absorption rate $R_T(E)$ is developed as follows:

$$R_T(E,T) = 1 - \prod_{Ag,Ta} \exp(-nl\sigma_T(E',T)) * F(E').\qquad(5)$$

According to the model given by Eq. (5), we analyse the absorption rate $R_{exp}(E)$ measured for different target temperature $T$, as shown in Fig. 3a.

Although the Ag target was kept at room temperature, it can be seen that the resonance widths of $^{109}$Ag vary from shot to shot. This is due to the fluctuation of $F(E)$ caused by statistical processes including neutron scattering on the beamline. Therefore, the $^{109}$Ag resonance serves as a reference to evaluate $F(E)$ for each measurement. In Fig. 3b, we plot the Doppler width $\Gamma_D(T)$ obtained by fitting $R_{exp}(E)$ as a function of $T$. The error bars are determined by the least mean square of the

difference between the fitted curves and experimental data. The Doppler width increases as the square root of $T$. We show a calculated result using a free gas model by Bethe[3]:

$$\Gamma_D(T) = 2\sqrt{\frac{m}{M}E_r k_B T},\qquad(6)$$

where $m$ and $M$ are the masses of the neutron and the target nucleus, respectively, and $k_B$ is Boltzmann's constant [the red dashed line in Fig. 3b]. We also introduce a proportionality coefficient $B$ in the right side of Eq. (6) to better reproduce the measured widths. The function obtained by least-square fitting is presented by the red solid line in Fig. 3b. This model can well reproduce the experimental results.

## Discussion
The temperature dependence of NRA is experimentally demonstrated by using single shot of our LDNS. This result shows that the

temperature of an element or an isotope inside of a material can be measured using NRA with a single neutron pulse of ~100 ns provided from a LDNS. Thermographic technologies such as X-ray interfero-metric imaging[1] and laser-based thermographic technique[2] have been studied. Although the present method requires a high-power laser system, it has the following two advantages. First, the temperature in a deep position can be measured compared to the laser technique[2]. Second, it is possible to measure selectively the temperature of an element in an object including different elements as demonstrated in the present study.

The temperature uncertainty is ~30 K around 600 K. Although the required resolution depends on the applications, this result suggests that the present method may be useful for various applications. The uncertainty is attributed to the systematical and statistical uncertain-ties. The systematical uncertainty can be improved by optimization of the TOF beamline. To decrease the statistical uncertainty, it is expec-ted to measure several times for a material under a same condition or to increase the neutron yield. The shortness of the moderated neutron pulse contributes to shortening of the neutron beamline with the high energy resolution in the TOF method, which enhances the number of neutrons arriving at a detector. Pumping laser systems operating at 10–100 Hz[37,38] will further improve the repetition rate, which indicates that real-time isotope-sensitive thermometry at the same frequency of the laser.

## Methods

### Neutron generation and moderation

The experiment was carried out using 1.5 ps laser pulses provided from Fast Ignition Experiment (LFEX) laser system[39], delivering a total energy of 900 J on the target at the Institute of Laser Engi-neering (ILE), Osaka University. The setup (Fig. 1) includes a vacuum chamber. A laser pulse with an intensity of ~$1 \times 10^{19}$ W·cm$^{-2}$ was focused on a 5 $\mu$m thick deuterated polystyrene (CD) foil to accel-erate protons and deuterons[22]. Before NRA measurement, the energies of the ions were measured by a Thomson parabola (TP) spectrometer[40] located in the normal direction of the CD foil. The protons and deuterons were accelerated up to MeV energy, where the ps duration laser pulse enhanced the acceleration of the deuterons[22]. To generate neutrons, the ions were injected into a secondary target, a Be cylinder with a diameter of 10 mm and a thickness of 10 mm. The diameter is larger than the angular diver-gence of the incident ions. The thickness is sufficient to stop the ions. Fast neutrons with energy over tens of MeV were generated via the $^9$Be(p,n)$^9$B and $^9$Be(d,n)$^{10}$B nuclear reactions. The maximum energy of the fast neutrons reaches 25 MeV, with a neutron number of $2.3 \times 10^{10}$ n/sr[22]. The total number of neutrons in the $4\pi$ direction reaches $10^{11}$ in a single laser shot[22].

### NRA measurements

For the temperature changing within 300–1000 K, sub-eV Doppler broadening width could be well investigated by NRS in eV energy region. Therefore, we located a neutron moderator on the LDNS to generate eV neutrons. The moderator was made from high-density polyethylene (0.98 g·cm$^{-3}$) in the shape of a cylinder and was attached to the Be target. The moderated neutrons passed through an aluminum window into air and their energies were measured at 1.78-m distance by a epithermal neutron TOF detector in which a $^6$Li-doped, Ce-activated glass scintillator (GS20, Scintacor Ltd., 10 mm in thickness) coupled to a time-gated PMT developed based on a HAMAMATSU-R2083[41]. To protect the PMT and ensure the linearity of the output response, the time gate was set to block the electro-magnetic pulses and the intense flash of X-rays generated by the laser-plasma interactions. To reduce the background due to neu-trons scattered by the chamber wall, we installed an nickel (Ni) col-limator on the beamline.

### Temperature depended resonance Doppler broadening measurements

We set a Ag metallic foil and a Ta metallic foil in the beamline to measure NRA of $^{109}$Ag at 5.19 eV and $^{181}$Ta at 4.28 eV. The thicknesses of the Ag and Ta targets are 0.2 mm and 0.1 mm, respectively. These materials are well-suited for demonstration purposes because their resonance peaks exhibit a large reaction cross sections and are not affected by nearby resonances. The Ta target was heated to $T = 361, 413, 474, 573$ and 617 K. The temperature was monitored by a thermocouple (Fig. 1). The Ag sample was kept at room temperature to provide a reference of the neutron spectrum. The NRA was measured for each temperature.

### Evaluation of neutron pulse temporal structure

To evaluate the temporal structure of the moderated neutron pul-ses, we used the Monte-Carlo simulation code–PHITS[42]. In the simulation, the neutrons are monitored at the exit of the moderator. The time duration of the moderation process is not negligible rela-tive to the total flight time to the detector. Supplementary Fig. 1 shows the temporal structure of the moderated neutrons $M(t)$ with kinetic energies 5.19 eV and 4.28 eV, which correspond to the reso-nance peak energies of $^{109}$Ag and $^{181}$Ta, respectively. Here, time zero indicates the time when the laser is first incident on the target, and the time for neutron generation in the LDNS is shorter than 50 ps[15–17,22]. The pulse broadening is evaluated to be 562 and 618 ns at the FWHM for 5.19 and 4.28 eV neutrons, respectively, based on the temporal structures [Supplementary Fig. 1a]. According to Ikeda-Carpenter function[43], the temporal duration of a moderated epi-thermal neutron pulse is proportional to $1/E^{0.5}$. By using the PHITS Monte-Carlo simulation code, we found that the pulse duration at the exit of the moderator [Supplementary Fig. 1a] is broadened as the $1/E^{0.5}$ rule. This broadening effect is involved in the present analysis.

The pulses are also broadened by the transit time of neutrons in the scintillator since the thickness of the scintillator (1 cm) is not negligible to the short beamline (1.78 m) in our experiment. The decay time of $^6$Li scintillation light is ~100 ns (FWHM), as reported in previous studies[44,45]. The pulse broadening at scintillator $S(t)$ con-sidering both scintillation and neutron transit time is evaluated to be 320 ns for 5 eV neutrons, shown as a red line in Supplemen-tary Fig. 1b.

By convolving the $M(t)$ [green line in Supplementary Fig. 1b] and the $S(t)$ [red line in Supplementary Fig. 1b], the total pulse broad-ening effect in the neutron detection is obtained as $D(t) = (M*S)(t)$ [blue line in Supplementary Fig. 1b], where * represents the opera-tion of convolution. By using the $D(t)$ function, we obtained TOF signals with the pulse broadening effects at the moderator and the detector.

## Data availability

The data that support the findings of this study are available on request from the corresponding authors.

## Code availability

The code that support the findings of this study are available on request from the corresponding authors.

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

## Acknowledgements

This work was funded by JSPS KAKENHI Grant-in-Aid for Scientific Research (JP25420911, JP26246043, JP22H02007, JP22H01239), JST A-STEP (AS2721002c) and JST PRESTO (JPMJPR15PD). Z.L. was supported by JSPS Research Fellowship for Young Scientists DC2 (202311207). Z.L. (before Apr. 2023) and T.W. were supported by JST SPRING, Grant Number JPMJSP2138. The authors thank the technical support staff of ILE for their assistance with the laser operation, target fabrication, and plasma diagnostics. This work was supported by the Collaboration Research Program of ILE, Osaka University. This work was partially supported by "Power Laser DX Platform" as research equipment shared in the Ministry of Education, Culture, Sports, Science and Technology Project for promoting public utilization of advanced research infrastructure (JPMXS0450300021).

## Author contributions

A.Y. provided overall supervision of the work. Z.L., Y.Ar., A.M., T.H., T.W., Y.T., Y.Ab., and A.Y. performed the experiment. Z.L., H.S., T.K., and M.K. analyzed the experimental data. S.F. designed the target. S.R.M. and K.M. reviewed and commented on the paper from theoretical viewpoints. A.Y. and R.K. managed the project. The manuscript was prepared by Z.L., T.H., and A.Y. All authors contributed to discussions and the preparation of the manuscript.

## Competing interests

The authors declare no competing interests.
