## [Peer Review File · Nature Communications]

Single-Shot Laser-Driven Neutron Resonance Spectroscopy for Temperature ProfilingReviewer #1 (Remarks to the Author):

The manuscript submitted to Nature Communication reports about the measurements of the temperature in a silver (Ag-109) and tantalum (Ta-181) by means of the broadening induced by the thermal motion of the crystal lattice to the zero temperature line width of a resonance, well described by the Breit - Wigner dispersion formula.

In principle there is nothing new about that, as the effect of the temperature on the resonance is well established, while the application to the possible measurement of the temperature of the interior of a material, selecting also elements is interesting.

Nevertheless, I think that the paper cannot be published in the present form as there are some issues that are not clear, nor well discussed and necessitate to be clarified and deepened.

One comment is about the thickness of the foils (100 μm for Ta-181 and 200 μm for Ag). This parameter is not important for itself but in relation the discussion about the lineshapes of the resonances discussed in figure 2(c). It is said that the saturation effect for the Ag signal comes from the effect $F(t)$ that "originates from neutron scattering in the beam line from the moderator". This explanation should be justified. If one considers the value of the resonance cross section for Ag-109 (about 30000 b or so) the signal might be completely saturated at $E=E_r$ (5.2 eV) so that absorption on the wings becomes important. Moreover the neutron spectrum in the epithermal region decays as $1/E^a$ (a being around unity) as indicated by the Ikeda Carpenter function. This $1/E^a$ should be convoluted to the pulse width at the given resonance or in the resonance regions (let say within the FWHM).

Another comment related to the possibility to use a single laser pulse. I refer to figure 2(a). There two issues that should be addressed: the first is that in order to define correctly the FWHM and extract the temperature from the doppler broadening the baseline should be well defined. In resonance analysis this issue is addressed by the technique of the so-called black resonances. Using very thick foil one is capable to provide an experimental measurement of the background beneath the resonance region. After that one can use a thinner foil so to measure the resonance parameters. Always in figure 2(a) I see that the two resonances overlaps and the interference in the wings regions (see figure 2(c)) should be discussed as this interference may determine large uncertainties in the determination of the FWHM. Does the use of a single laser pulse provide the due statistical accuracy to apply this procedure if this is needed for the application discussed in the paper? I suppose so, but authors should clearly addressed this point.

As matter of fact, figure 3(b) shows the trend of the doppler width as a function of T. Especially the two last points and the two points around 400 K have compatible values within the 1sigma associated uncertainty and this "temperature resolution" should be commented assessing to what extent is reliable for the use the authors intend to pursue.

A few comments about the text:

- 1- please quote the values of the peak cross sections for Ag-109 and Ta-181
- 2- the emission of the pulsed neutrons in a pulsed source (for example a spallation source) is not spherical as the forward neutron spectrum (i.e. around the momentum of the incident accelerated particle) is different with respect to the backward direction.
- 3- I ask authors to pay attention to the English language over the whole text.

Standing my previous comments, I do not recommend the publication of the paper in the present form and major revision is needed.

Reviewer #2 (Remarks to the Author):

The manuscript follows previous works of different authors on temperature measurements of isotopes using NRA. Previous works, like the one mentioned by the authors in ref. [8] (A. Tremsin, et al. 2015), clearly show the possibility of temperature measurement via NRA by using conventional neutron sources. In this work, the authors investigate the possibility of doing the same using laser-driven neutron, as it has the advantage of a shorter Time-of-flight, that is a smaller beamline, and a very short time exposure, that is a single laser shoot.

This type of approach for temperature measurement can be useful in several applications or fundamental research. To my knowledge, the work is new and matches the requirements of Nat. Comm.

Although I find the subject of this work generally worth to be published, it seems that the manuscript has not been carefully edited by the authors. I have found several inaccuracies and a lack of clarity in some parts. Generally, the manuscript needs to be improved in clarity and rigorousness and then resubmitted.

The math is not rigorous and, in some cases, makes the expressions not clear.

For instance, although the authors defined in the Methods the \otimes symbol as the convolution operator, usually this operator is the Tensor product, while the convolution is commonly defined with the following symbol $*$. It would be better to use the latter symbol then.

At page 4, in the section 'Results' at line 8, the word 'deconvoluted' is misspelled as 'deconvoluted'. Also, in the expression of the ToF, $t_{\text{RAW}} = t \otimes D(t)$, the operator might be replaced with $*$, as mentioned above, that is, $t_{\text{RAW}} = (t * D)(t)$.

The definition of $R_{\text{exp}}(E)$ at page 4 is not clear. It has been defined as the experimental neutron absorption rate, in line with the definition of $R_0(E)$ as the theoretical absorption rate. However, if one looks at the definition of $R_0(E)$ in Eq.1, and presumably considers the cross-section $\sigma(E)$ to be the one represented in Fig. 2b, it is not possible that the absorption rate $R_0(E)$ saturates around the resonances 4.28 and 5.19 eV, as reported in Fig.2c (Also, $R_0(E)$ is mentioned to be illustrated in Fig. 2b, but probably it is in Fig.2c). Actually, oppositely, if the thickness of the material increases the value of the $R_0(E)$ decreases.

Although, in the Fig.2c the y-axis refers to 'neutron absorption' instead of 'neutron absorption rate'. These are two different things. In fact, if Fig.2c represents the neutron absorbed by the material, then if there are not enough neutrons, indeed the absorption can saturate, in the sense that all neutrons in a given range of energies can get fully caught by the material.

This part is very confusing and needs clarity.

Regarding Eq.1, the authors define the density 'n' as the atomic areal number density, while it is a proper volumetric density.

Also, the authors say that the cross-section $\sigma(E)$ was obtained via JENDL4.0 for a temperature equal to room temperature without mentioning the actual value of the temperature used. I presume the cross-section is illustrated in Fig.2b, but this is not mentioned in the manuscript, as in Fig. 2b the temperature is not mentioned too.

At page 5 line 5, the authors say that the difference between $R_0(E)$ and $R_{\text{exp}}(E)$ indicates the presence of another effect, $F(t)$, that causes further pulse broadening in addition to $D(t)$. Although this may be true, the authors consider $F(t)$ to originate from statistical neutron scattering in the beamline, while part of the broadening must necessarily come from the Doppler effect. It seems that the authors want to consider the effect of $F(t)$ to be decoupled from the Doppler effect and define Eq.2 ($R_1(E) = R_0(E) \otimes F(E)$) as the Eq. to use to account for the effect of $F(t)$ and $D(t)$. This is in my opinion wrong, and in fact also contradicting what the authors correctly say later, that Eq.5 is used to fit the experimental curves of Fig.3 (see caption of Fig.3a). The text is not clearly written.

In Fig2a the y-axis writes "Oscillator signal", while I suppose the authors report an "Oscilloscope signal".

The cross-section for Ag of Fig.2b, which supposedly is at room temperature, is about a factor 2 higher than the one reported in Fig.2d at 300k (i.e., room temperature). It is not clear why.

The legend of Fig.2d seems wrong, as the JENDL data and BW model temperatures seem swapped. Please clarify.

At page 5, line 5 of the section "Temperature dependence on resonance Doppler broadening", the authors introduce the Breit-Wigner (BW) single-level formula for the neutron absorption cross-section, $\sigma_{BW}(E')$, which they say to be for a temperature of 0 K. However, the cross-section itself is independent of the temperature as the energy E' is the relative energy of the neutron with respect to a nucleus of the absorber. Also, they state that " $E' = E$ at 0 K", while it is not clear what E is. Please revise the statement to be clearer.

In Eq.4 the authors define the cross-section $\sigma_T(E)$ that takes into account the Doppler effect, however, the formula seems not to be correct.

The variable of the convolved function is E , that is $\sigma_T(E)$, while from Eq. it seems also that the integration variable is E .

Presumably, the integration variable should be the relative energy E' , both in the σ_{BW} and the Gaussian function.

Similarly, as above, in Eq.5, the integration variable and the variable of the convoluted function R_T cannot be the same. The variable in the expression " $\exp(-n\sigma_T(E)) \otimes F(E)$ " should be the relative energy E' .

In the formula in Eq.6, the authors report the Doppler width $\Gamma_D(T)$, saying that this formula is coming from the free gas model by Bethe in reference [3]. However, ref[3] doesn't report that formula, as probably the sum between the mass of the neutron, m , and the nucleus, M , is approximated to the mass of the nucleus as much heavier.

Please, write also the reference from where you took the formula that you show in Eq.6.

Also, Eq.6 is wrong as the number 2 needs to be outside of the radical.

Furthermore, I do not understand the symbol of proportionality \propto for Eq.6 as if one doesn't know the exact expression for Doppler width $\Gamma_D(T)$ then one cannot retrieve the temperature. This is at the core of the retrieval of the temperature so it needs an explanation.

Reviewer #3 (Remarks to the Author):

The work demonstrates the ability to determine the temperature of material even if it is embedded in a complicated environment. For this the position of the neutron resonance line serves to select a certain isotope, and in addition then the temperature dependence on resonance Doppler broadening to determine the temperature. The position and width of the neutron resonance lines is measured via time of flight analysis of the transmitted neutrons.

The ability to gain data accurate enough to reach an accurate temperature determination requires both a short emission time and a sufficient statistical quality of the data. For this the laser driven neutron source (LDNS) is chosen. Since the resonance absorption lines of interest are at relatively low neutron energies, in addition the moderation of the neutron energy has to maintain the time sharpness of the pulse.

The problems (and solutions) to reach these qualities are only shortly mentioned in this publication, but sufficiently described and also covered by the citations.

A key ingredient for the present work is the use of a very strong laser source (LFEX, 1.5 ps at 900 J). The authors show well resolved spectra of the ^{181}Ta resonance at $E_r = 4.28$ eV, and the ^{109}Ag resonance at $E_r = 5.19$ eV. The temperature of the tantalum is varied, while silver is kept at room temperature.

A key argument for the usefulness is the comparison of the spectra with the calculated result from the Breit-Wigner (BW) single-level formula, which is shown to be in well agreement with the given temperature of the target and the results.

In total the authors rightly claim a good agreement between the measured and predicted values, which make the method applicable for the measurement of parameters of complicated material mixtures like inertial fusion energy related plasmas.

A drawback of the presented set-up is the need for a very complicated laser system. Here the authors should explain the difference to other approaches, which are partly also cited in the paper. The result would be a claim to be able to deliver a useful method urgently needed.

Even under this restriction the paper seems to be very well suited for publication in Nature Communications.

Response to the reviewer#1

First of all, the authors deeply appreciate the fruitful comments by the reviewer, which are completely fair and quite useful to improve our manuscript. The authors guess that the reviewer is truly an outstanding specialist on neutron science. We show the revised part as **bold** in the manuscript and list the response to the reviewer's comments below:

Reviewer's comment:

Reviewer #1 (Remarks to the Author):

The manuscript submitted to Nature Communication reports about the measurements of the temperature in a silver (Ag-109) and tantalum (Ta-181) by means of the broadening induced by the thermal motion of the crystal lattice to the zero temperature line width of a resonance, well described by the Breit - Wigner dispersion formula.

In principle there is nothing new about that, as the effect of the temperature on the resonance is well established, while the application to the possible measurement of the temperature of the interior of a material, selecting also elements is interesting.

Nevertheless, I think that the paper cannot be published in the present form as there are some issues that are not clear, nor well discussed and necessitate to be clarified and deepened.

One comment is about the thickness of the foils (100 μm for Ta-181 and 200 μm for Ag). This parameter is not important for itself but in relation the discussion about the line shapes of the resonances discussed in figure 2(c). It is said that the saturation effect for the Ag signal comes from the effect $F(t)$ that "originates from neutron scattering in the beam line from the moderator". This explanation should be justified. If one considers the value of the resonance cross section for Ag-109 (about 30000 b or so) the signal might be completely saturated at $E=E_r$ (5.2 eV) so that absorption on the wings becomes important.

Response to the reviewer:

As the reviewer commented, the saturation effect around the Ag peak in Fig. 2(a) is caused by the thick target that can absorb almost all neutrons around the resonance peak of Ag with the large neutron capture cross section. As explained later, we determine the background level for each laser shot using the absorption peak depth of the thick Ag target. In addition to the thick target effect, we have introduced the $F(t)$ as an additional broadening effect during flight of the neutrons.

- Page 4-6, Line 101-110

Newly added descriptions in the revised article:

In present analysis, we evaluate the background level for each laser shot using the reference foil (Ag) in the following procedure similar to black resonance [35]. The baseline [the blue dashed line in Fig. 2(a)] is obtained from a least-square fitting for the raw signals excluding the two resonance dips. The

background (the green solid line) is obtained from the baseline by subtracting the depth of the resonance peak of ^{109}Ag , because the Ag target is enough thick to absorb almost all neutrons around the resonance energy. From the difference between the law signal and the baseline, the experimental absorption rate $R_{\text{exp}}(E)$ normalized to the difference between the baseline and the background is obtained as the blue dashed line in Fig. 2(c).

Reviewer's comment:

Moreover the neutron spectrum in the epithermal region decays as $1/E^a$ (a being around unity) as indicated by the Ikeda Carpenter function. This $1/E^a$ should be convoluted to the pulse width at the given resonance or in the resonance regions (let say within the FWHM).

Response to the reviewer:

The pulse broadening due to the Ikeda-Carpenter effect has already involved in the analysis seen in the first version of the paper. As suggested by the reviewer, the neutron spectrum in the epithermal region decays as $1/E^a$ with $a = \sim 1$. In addition, according to Ikeda-Carpenter function [43], the temporal duration of a moderated epithermal neutron pulse is proportional to $1/E^{0.5}$. We have calculated pulse duration using the PHITS Monte-Carlo simulation assuming the experimental conditions. In this simulation, the scattering of neutrons on the target and moderator have been considered and the energy spectrum has been obtained. The obtained energy spectrum corresponded to a $1/E$ spectrum. As shown in Fig. 4(a), the pulse durations are 618 ns at 4.28 eV and 562 ns at 5.19 eV. From the value at 5.19 eV and the energy difference, we obtain the expected width of $562 \text{ ns} \times (5.19 \text{ eV}/4.28 \text{ eV})^{0.5} = 618.87 \text{ ns}$ at 4.28 eV. In this way, the pulse duration is broadened according to the $1/E^{0.5}$ formula. Hence, we explain it in the revised paper as follows:

- Page 12-13, Line 251-255

Newly added descriptions in the revised article:

According to Ikeda-Carpenter function [43], the temporal duration of a moderated epithermal neutron pulse is proportional to $1/E^{0.5}$. By using the PHITS Monte-Carlo simulation code, we found that the pulse duration at the exit of the moderator [Fig. 4(a)] is broadened as the $1/E^{0.5}$ rule. This broadening effect is involved in the present analysis.

- Page 18

We added the reference:

[42] S. Ikeda and J. M. Carpenter, Nuclear Instruments and Methods in Physics Research Section A: Accelerators, Spectrometers, Detectors and Associated Equipment 239, 536 (1985).

Reviewer's comment:

Another comment related to the possibility to use a single laser pulse. I refer to figure 2(a). There two issues that should be addressed: the first is that in order to define correctly the FWHM and extract the temperature from the doppler broadening the baseline should be well defined. In resonance analysis this

issue is addressed by the technique of the so-called black resonances. Using very thick foil one is capable to provide an experimental measurement of the background beneath the resonance region. After that one can use a thinner foil so to measure the resonance parameters.

Response to the reviewer:

Since the neutrons and backgrounds of LDNS could be obviously changed in every shot. Using a thick target (Ta) to measure the baseline in a single shot cannot be referred in other shots. This results in the original black resonance method not being applicable to LDNS. In the present method, we use a reference material for the same laser shot. This reference can be also used for evaluation of the background. As explained previously for the first question of the reviewer, the reference is enough thick to absorb almost all neutrons at the resonance energy of the reference material. Because the resonance energies of the reference and the sample to be measured are close to each other, we can evaluate the background at the resonance energy of the sample material as shown in the green line in Fig. 2(a). This approach is similar with the black resonance method. First, we determine the baseline from the raw signal by least-square fitting excluded the two resonance dips. As explained above, the reference target of Ag is enough thick to absorb almost all neutrons at the resonance peak. Thus, we can obtain the background level. Next, we obtained the background from the baseline by subtracting the depth of the dip of the resonance of Ag. Third, the difference is normalized to the difference between the baseline and the background. To clarify this point, we added some sentences as below.

- Page 4-6, Line 101-110

As explained previously, we newly added descriptions in the revised article:

In the present analysis, we evaluate the background level for each laser using the reference foil (Ag) in the following procedure similar to black resonance [35]. The baseline [the blue dashed line in Fig. 2(a)] is obtained from a least-square fitting for the raw signals excluding the two resonance dips. The background (the green solid line) is obtained from the baseline by subtracting the depth of the resonance peak of ^{109}Ag around $54\ \mu\text{s}$ in Fig. 2(a), because the Ag target is enough thick to absorb almost all neutrons around the resonance energy. From the difference between the raw signal and the baseline, the experimental absorption rate $R_{\text{exp}}(E)$ normalized to the difference between the baseline and the background is obtained as the blue dashed line in Fig. 2(c).

Reviewer's comment:

Always in figure 2(a) I see that the two resonances overlaps and the interference in the wings regions (see figure 2(c)) should be discussed as this interference may determine large uncertainties in the determination of the FWHM. Does the use of a single laser pulse provide the due statistical accuracy to apply this procedure if this is needed for the application discussed in the paper? I suppose so, but authors should clearly addressed this point.

Response to the reviewer:

As pointed by the reviewer, the two peaks in Fig. 2(a) are a bit close to each other and there is an overlapped wing region.

As for the FWHM analysis, we used the fitting function considering the overlapped lineshape of the two resonances by multiplying the two absorption rates [please check Eq. (5)]. After the fitting, the FWHM is evaluated with the resonance peak of Ta which is individually calculated with the parameter obtained from fitting. For example, please see Fig. 3(a) in Page 9. The red line is total absorption rate fitting and the blue line is re-calculated Ta resonance absorption.

In future applications, we should choose more carefully the reference material which the resonance energy is enough far from the resonance energy of the material to be measured.

We add the discussion and explanation as below.

- Page 6, Line 131-136

Although the two resonance peaks are close to each other in the present experiment, this overlapping region was already considered in the FWHM analysis as explained later. When we chose the reference material whose resonance energy is enough far from the resonance energy of the material to be measured, the uncertainty caused from the overlapping between the two resonances becomes negligibly small.

Reviewer's comment:

As matter of fact, figure 3(b) shows the trend of the doppler width as a function of T. Especially the two last points and the two points around 400 K have compatible values within the 1sigma associated uncertainty and this "temperature resolution" should be commented assessing to what extent is reliable for the use the authors intend to pursue.

Response to the reviewer:

Because the required temperature resolution depends on applications, it is, in general, difficult to discuss it. The uncertainty of the measured temperature is about 30 K around 600 K, corresponding to $dT/T=5\%$. We expect that this temperature uncertainty may be useful for various applications. The measurement uncertainty was predominantly attributed to the symmetrical uncertainty from the TOF data uncertainty and the statistical uncertainty leading the fitting uncertainty. One of the methods to improve the total uncertainty is to measure a material several times for the same condition so that the uncertainty originated from the statistics can be improved. Furthermore, to decrease the systematical uncertainty, the development on beamlines and neutron sources is important. We add a belief discussion for the temperature uncertainty.

- Page 9-10, Line 191-197

The temperature uncertainty is approximately 30 K around 600 K. Although the required resolution depends on applications, this result suggests that the present method may be useful for various applications. The uncertainty is attributed to the systematical and statistical uncertainties. The systematical uncertainty can be improved by optimization of the TOF beamline. To decrease the statistical uncertainty, it is expected to measure several times for a material under a same condition or to increase the neutron yield.

Reviewer's comment:

A few comments about the text:

1- please quote the values of the peak cross sections for Ag-109 and Ta-181.

Response to the reviewer:

The peak cross sections for Ag-109 and Ta-181 were added in Page 4.

- Page 4, Line 98-100

The neutron absorption cross sections at 300 K are approximately 12500 and 23000 barns at 4.28 eV (¹⁸¹Ta) and 5.19 eV (¹⁰⁹Ag), respectively.

Reviewer's comment:

2- the emission of the pulsed neutrons in a pulsed source (for example a spallation source) is not spherical as the forward neutron spectrum (i.e. around the momentum of the incident accelerated particle) is different with respect to the backward direction.

Response to the reviewer:

The authors sincerely apologize for using such wrong description. The text has been modified as below.

- Page 3, Line 53-54

Previous article:

However, a long beamline reduces the flux of neutrons arriving at a detector due to the spherically emission of pulsed neutron sources.

Revised article:

However, a long beamline reduces the flux of neutrons arriving at a detector due to the smaller solid angle of the detector.

Reviewer's comment:

3- I ask authors to pay attention to the English language over the whole text.

Response to the reviewer:

We apologize for causing such unnecessary trouble in reading. The English over the whole text has been checked and the expression that we thought was most appropriate was used.

Response to the reviewer#2

First of all, the authors deeply appreciate the fruitful comments by the reviewer, which are completely fair and quite useful to improve our manuscript. The responses to the comments are listed below. All the revisions are shown in **bold** font in the manuscript.

Reviewer's comment:

Reviewer #2 (Remarks to the Author):

The manuscript follows previous works of different authors on temperature measurements of isotopes using NRA. Previous works, like the one mentioned by the authors in ref. [8] (A. Tremsin, et al. 2015), clearly show the possibility of temperature measurement via NRA by using conventional neutron sources. In this work, the authors investigate the possibility of doing the same using laser-driven neutron, as it has the advantage of a shorter Time-of-flight, that is a smaller beamline, and a very short time exposure, that is a single laser shoot.

This type of approach for temperature measurement can be useful in several applications or fundamental research. To my knowledge, the work is new and matches the requirements of Nat. Comm.

Although I find the subject of this work generally worth to be published, it seems that the manuscript has not been carefully edited by the authors. I have found several inaccuracies and a lack of clarity in some parts. Generally, the manuscript needs to be improved in clarity and rigorousness and then resubmitted.

The math is not rigorous and, in some cases, makes the expressions not clear. For instance, although the authors defined in the Methods the \otimes symbol as the convolution operator, usually this operator is the Tensor product, while the convolution is commonly defined with the following symbol $$. It would be better to use the latter symbol then.*

At page 4, in the section 'Results' at line 8, the word 'deconvoluted' is misspelled as 'deconvolued'. Also, in the expression of the ToF, $t_{RAW} = t \otimes D(t)$, the operator might be replaced with $$, as mentioned above, that is, $t_{RAW} = (t * D)(t)$.*

Response to the reviewer:

We thank the reviewer 2 for providing the correct mathematical expression. The function in Page 4 and the operator of convolution have been modified as the comments. We apologize for the misspelling of the word 'deconvolued'. The word has been fixed to '**deconvoluted**'.

Reviewer's comment:

The definition of $R_{exp}(E)$ at page 4 is not clear. It has been defined as the experimental neutron absorption rate, in line with the definition of $R_0(E)$ as the theoretical absorption rate. However, if one looks at the definition of $R_0(E)$ in Eq.1, and presumably considers the cross-section $\sigma(E)$ to be the one represented in Fig. 2b, it is not possible that the absorption rate $R_0(E)$ saturates around the resonances 4.28 and 5.19 eV, as reported in Fig.2c (Also, $R_0(E)$ is mentioned to be illustrated in Fig. 2b, but probably it is in Fig.2c). Actually, oppositely, if the thickness of the material increases the value of the $R_0(E)$ decreases.

Response to the reviewer:

As shown in Fig. 2(c), the absorption rates $R_0(E)$ are the number ratio of the absorbed neutrons to the incident neutrons after penetration of a target. When all neutrons are absorbed, the ratio becomes 1 (or 100%). We apologize for the incorrect formula of Eq. (1) in the original manuscript. The correct formula is

$$R_0(E) = 1 - \prod_{Ag, Ta} \exp(-nl\sigma(E)),$$

for multi targets. Note that the sum operator ' Σ ' in Eq. (1) and Eq. (5) in the original manuscript should be product ' \prod '. As the thickness of the material increases, the value of the $R_0(E)$ increases.

The absorption rates $R_0(E)$ around the resonance peak show the saturation for the thick Ag target as shown in Fig. 2(c).

Reviewer's comment:

Although, in the Fig.2c the y-axis refers to 'neutron absorption' instead of 'neutron absorption rate'. These are two different things. In fact, if Fig.2c represents the neutron absorbed by the material, then if there are not enough neutrons, indeed the absorption can saturate, in the sense that all neutrons in a given range of energies can get fully caught by the material. This part is very confusing and needs clarity.

Response to the reviewer:

Thank the reviewer for this comment. We have modified the title of y-axis in Fig. 2(c) as **Neutron absorption rate**.

Reviewer's comment:

Regarding Eq.1, the authors define the density 'n' as the atomic areal number density, while it is a proper volumetric density.

Response to the reviewer:

As the reviewer commented, the 'n' is the volumetric number density instead of the areal number density. We have modified the mistake.

- Page 6, Line 113, following the Eq. (1)

where n is the volumetric number density, ...

Reviewer's comment:

Also, the authors say that the cross-section $\sigma(E)$ was obtained via JENDL4.0 for a temperature equal to room temperature without mentioning the actual value of the temperature used. I presume the cross-section is illustrated in Fig.2b, but this is not mentioned in the manuscript, as in Fig. 2b the temperature is not mentioned too.

Response to the reviewer:

We have modified the manuscript as below.

- Page 5, in the caption of Fig. 2(b)

(b) The neutron absorption cross sections of ^{109}Ag and ^{181}Ta **at 300 K**, obtained from JENDL4.0 data base [34].

- Page 5, in the Fig. 2(b)

We added the temperature in the Fig. 2(b).

Reviewer's comment:

At page 5 line 5, the authors say that the difference between $R_0(E)$ and $R_{exp}(E)$ indicates the presence of another effect, $F(t)$, that causes further pulse broadening in addition to $D(t)$. Although this may be true, the authors consider $F(t)$ to originate from statistical neutron scattering in the beamline, while part of the broadening must necessarily come from the Doppler effect. It seems that the authors want to consider the effect of $F(t)$ to be decoupled from the Doppler effect and define Eq.2 ($R_1(E) = R_0(E) \otimes F(E)$) as the Eq. to use to account for the effect of $F(t)$ and $D(t)$. This is in my opinion wrong, and in fact also contradicting what the authors correctly say later, that Eq.5 is used to fit the experimental curves of Fig.3 (see caption of Fig.3a). The text is not clearly written.

Response to the reviewer:

Thank the reviewer 2 for his/her comments on this part. The cross section $\sigma(E)$ used in $R_0(E)$ is the cross section data **at 300 K**, obtained from JENDL4.0. The Doppler effect at this temperature has been already considered in the lineshape of the cross section data. Therefore, we define the $F(E)$ to explain the addition broadening.

In Eq. (3), we used the original Breit-Wigner single-level formula to describe the neutron absorption cross section at **0 K**, as $\sigma_{BW}(E)$. Then, in Eq. (4), we present the temperature dependent part including Doppler width $\Gamma_D(T)$ to obtain the cross section at the sample's temperature. Finally, we obtained the Eq. (5) from Eq. (3) and Eq. (4).

- Page 6, Line 113,

..., **l is the thickness of the targets and $\sigma(E)$ is the NRA cross section (JENDL4.0 [34]) at 300 K [Fig. 2(b)].**

Reviewer's comment:

In Fig2a the y-axis writes "Oscillator signal", while I suppose the authors report an "Oscilloscope signal".

Response to the reviewer:

We agree with the reviewer. We changed the title of Y-axis in Fig. 2(a) to "Oscilloscope signal".

Reviewer's comment:

The cross-section for Ag of Fig.2b, which supposedly is at room temperature, is about a factor 2 higher than the one reported in Fig.2d at 300k (i.e., room temperature). It is not clear why.

Response to the reviewer:

That is a mistake. We corrected the cross-section data and y-axis of Fig. 2(d).

Reviewer's comment:

The legend of Fig.2d seems wrong, as the JENDL data and BW model temperatures seem swapped. Please clarify.

Response to the reviewer:

As pointed out by the reviewer, the legend of Fig. 2(d) was incorrect in the previous manuscript. We fixed them.

Reviewer's comment:

At page 5, line 5 of the section "Temperature dependence on resonance Doppler broadening", the authors introduce the Breit-Wigner (BW) single-level formula for the neutron absorption cross-section, $\sigma_{BW}(E')$, which they say to be for a temperature of 0 K. However, the cross-section itself is independent of the

temperature as the energy E' is the relative energy of the neutron with respect to a nucleus of the absorber. Also, they state that " $E' = E$ at 0 K", while is not clear what E is. Please revise the statement to be clearer.

Response to the reviewer:

As the reviewer's comment, the expression of E' and E is confusing. We have changed E' in Eq. (3) to E as below.

- Page 7

$$\sigma_{BW}(E) = \pi \lambda g_j \frac{\Gamma_n \Gamma_\gamma}{(E - E_r)^2 + (\Gamma_n + \Gamma_\gamma)^2/4},$$

And the explanation following Eq. (3) has been revised as:

- Page 7, line 145

where E_r is the resonance energy, **E is the kinetic energy of the incident neutron,**...

Reviewer's comment:

In Eq.4 the authors define the cross-section $\sigma_T(E)$ that takes into account the Doppler effect, however, the formula seems not to be correct.

The variable of the convolved function is E , that is $\sigma_T(E)$, while from Eq. it seems also that the integration variable is E .

Presumably, the integration variable should be the relative energy E' , both in the σ_{BW} and the Gaussian function.

Response to the reviewer:

We have fixed the mistakes as below:

- Page 7, Eq. (4)

$$\sigma_T(E, T) = A \times \sigma_{BW}(E') * \exp\left(-\frac{(E' - E_r)^2}{2\Gamma_D^2(T)}\right),$$

- Page 7, Line 157-159, following the Eq. (4)

where A is a fitting parameter and E' is the relative neutron energy for integration variable. σ_{BW} and the following Doppler broadening term are convoluted in the energy axis. The Doppler width $\Gamma_D(T)$ broadens with the temperature T .

Reviewer's comment:

Similarly, as above, in Eq.5, the integration variable and the variable of the convoluted function R_T cannot be the same. The variable in the expression “ $\exp(-nI\sigma_T(E)) \otimes F(E)$ ” should be the relative energy E' .

Response to the reviewer:

We have fixed Eq. (5) as below:

- Page 7, Eq. (5)

$$R_T(E, T) = 1 - \prod_{A_g, T_a} \exp(-nI\sigma_T(E', T)) * F(E')$$

Reviewer’s comment:

In the formula in Eq.6, the authors report the Doppler width $\Gamma_D(T)$, saying that this formula is coming from the free gas model by Bethe in reference [3]. However, ref[3] doesn’t report that formula, as probably the sum between the mass of the neutron, m , and the nucleus, M , is approximated to the mass of the nucleus as much heavier.

Please, write also the reference from where you took the formula that you show in Eq.6. Also, Eq.6 is wrong as the number 2 needs to be outside of the radical.

Response to the reviewer:

We derived independently this formula from the Maxwell-Boltzmann distribution of velocity of nuclei. We consider that the well-known formula by Bethe is an approximation for $m \ll M$, and in this case the results using the two formulas are almost identical. Thus, it is more suitable for understanding of readers that we describe the Bethe formula instead of our formula.

- Page 9, Eq. (6), the formula has been modified to the reduced form.

$$\Gamma_D(T) = 2\sqrt{\frac{m}{M} E_r k_B T},$$

- Page 8, Fig. 3(b)

The formula of Γ_D in Fig. 3(b) has been modified to the reduced form.

Reviewer’s comment:

Furthermore, I do not understand the symbol of proportionality \propto for Eq.6 as if one doesn’t know the exact expression for Doppler width $\Gamma_D(T)$ then one cannot retrieve the temperature. This is at the core of the retrieval of the temperature so it needs an explanation.

Response to the reviewer:

In the previous manuscript, the widths displayed in Fig. 3(b) were the values of 2×FWHM for the Gaussian function that is obtained by fitting with Eq. (5) for each neutron resonance. There is a relationship of FWHM = 2.35× $\Gamma_D(T)$. Therefore, the widths (2×FWHM) in Fig. 3(b) are about 4.7 times of $\Gamma_D(T)$. Thus, the displayed widths are proportional to $\Gamma_D(T)$. Furthermore, we introduced a factor B to better reproduce the widths (2×FWHM) by a least-square fitting as

$$\Gamma_D(T) = B \times 2 \sqrt{\frac{m}{M} E_r k_B T}.$$

This is the reasons that we used the symbol of ‘ α ’ in the previous manuscript. However, as pointed out by the reviewer, this was confusing. In the Fig. 3(b) of the revised manuscript, we changed from the values of (2×FWHM) to the experimental $\Gamma_D(T)$ for the widths. We newly added that the exact function using Eq. (6) as the red dashed line. This line is slightly shifted from the experimental $\Gamma_D(T)$. And we displayed still another function shown in the above equation (the red solid line), where the parameter B is determined by least-square fitting to reproduce the experimental $\Gamma_D(T)$.

- Page 8, Fig. 3(b)

The figure was fixed with the unified data.

- Page 9, Line 175-178

Previous article:

..., and k_B is Boltzmann’s constant.

Revised article:

..., and k_B is **Boltzmann’s constant [the red dashed line in Fig. 3(b)]. We also introduce a proportionality coefficient B in the right side of Eq. (6) to better reproduce the measured widths. The function obtained by least-square fitting is presented by the red solid line in Fig. 3(b).**

Reviewer’s comment:

Reviewer #3 (Remarks to the Author):

The work demonstrates the ability to determine the temperature of material even if it is embedded in a complicated environment. For this the position of the neutron resonance line serves to select a certain isotope, and in addition then the temperature dependence on resonance Doppler broadening to determine the temperature. The position and width of the neutron resonance lines is measured via time of flight analysis of the transmitted neutrons.

The ability to gain data accurate enough to reach an accurate temperature determination requires both a short emission time and a sufficient statistical quality of the data. For this the laser driven neutron source (LDNS) is chosen. Since the resonance absorption lines of interest are at relatively low neutron energies, in addition the moderation of the neutron energy has to maintain the time sharpness of the pulse.

The problems (and solutions) to reach these qualities are only shortly mentioned in this publication, but sufficiently described and also covered by the citations.

A key ingredient for the present work is the use of a very strong laser source (LFEX, 1.5 ps at 900 J).

The authors show well resolved spectra of the ^{181}Ta resonance at $E_r = 4.28$ eV, and the ^{109}Ag resonance at $E(r) = 5.19$ eV. The temperature of the tantalum is varied, while silver is kept at room temperature.

A key argument for the usefulness is the comparison of the spectra with the calculated result from the Breit-Wigner (BW) single-level formula, which is shown to be in well agreement with the given temperature of the target and the results.

In total the authors rightly claim a good agreement between the measured and predicted values, which make the method applicable for the measurement of parameters of complicated material mixtures like inertial fusion energy related plasmas.

Response to the reviewer:

The authors deeply appreciate the fruitful comments by the reviewer.

Reviewer's comment:

A drawback of the presented set-up is the need for a very complicated laser system. Here the authors should explain the difference to other approaches, which are partly also cited in the paper. The result would be a claim to be able to deliver a useful method urgently needed.

Even under this restriction the paper seems to be very well suited for publication in Nature Communications.

Response to the reviewer:

As commented by the reviewer, it is important to compare the LDNS with other approaches that cited in the article. To supplement this part, we add some sentences in the discussion as below.

- Page 9, Line 184-190

Thermographic technologies such as X-ray interferometric imaging [1] and laser-based thermographic technique [2] have been studied. Although the present method requires a high-power laser system, it has the following two advantages. First, the temperature in a deep position can be measured compared to the laser technique [2]. Second, it is possible to measure selectively the temperature of an element in an object including different elements as demonstrated in the present study.

Reviewer #1 (Remarks to the Author):

The revised version of the manuscript provides a greatly improved version of the original paper. Authors have replied to all my comments and provided the corrections where required.

I found the revised version of the manuscript worth to be published in the present form.

Reviewer #2 (Remarks to the Author):

The Authors have improved a lot the correctness and comprehensibility of the manuscript by making several amendments, as suggested by all Reviewers. I am fine with publishing the manuscript in its current form.

Response to the reviewers

The fruitful comments and discussions by the reviewers have drastically improved our manuscript. I would like to express our great appreciation to the reviewers.

Sincerely,

Akifumi Yogo

Ph.D., Professor

Institute of Laser Engineering, Osaka University,

2-6, Suita 565-0871, Osaka, Japan

Tel: +81 6 6879 8766, E-mail: yogo-a@ile.osaka-u.ac.jp